# Site History's Role in Urban Agriculture: A Case Study in Kisumu, Kenya, and Ouagadougou, Burkina Faso

Nicolette Tamara Jonkman [1],*, Karsten Kalbitz [2], Huig Bergsma [3] and Boris Jansen [1]

1    Ecosystem and Landscape Dynamics Group, Institute for Biodiversity and Ecosystem Dynamics, University of Amsterdam, 1090 GE Amsterdam, The Netherlands; b.jansen@uva.nl
2    Institute of Soil Science and Site Ecology, Technische Universität Dresden, 01737 Tharandt, Germany; karsten.kalbitz@tu-dresden.de
3    BodemBergsma, Blikakker 8, 7421 GD Deventer, The Netherlands
*    Correspondence: n.t.jonkman@uva.nl

**Abstract:** Urban agriculture (UA) is a widespread practice often considered low-profit, taking place on marginal lands. This is supported by the lack of quantitative data on UA's contributions to food security and employment, yet contradicted by prevalence and high participation rates. This case study of six UA sites in Kisumu, Kenya and Ouagadougou, Burkina Faso explores the relationship between prior land use and current management and soil quality. A soil survey is performed determining the soil macronutrient and soil mineral composition. Agricultural management, ownership, and prior land use are investigated through interviews, satellite imagery, and historic publications. Results show three UA sites predating surrounding urban development, and data on soil nutrient content show that sites likely were chosen for their soil. The three younger sites are smaller and less embedded in the local economy, but soil analysis shows medium-rich to rich agricultural soils. We conclude that one cannot assume that UA is practiced on marginalized soils. Consequently, both value attribution to and the sustainable agricultural management of UA soils must be based on their characteristics, such as mineralogy and nutrient status, to prevent valuable soil resources from being lost. Through this, the more accurate value attribution of UA can be achieved, lending weight to the value attributed to UA by local communities.

**Keywords:** urban agriculture; value attribution; land use history

## 1. Introduction

Urban agriculture (UA) is the practice of agricultural activities such as livestock keeping and vegetable growing in the urban environment [1,2]. Vegetable growing flourishes in many cities in sub-Saharan Africa (SSA) and Southeast Asia [2–6]. Different stakeholders take conflicting stances on the issue of UA in practice. Some consider that the practice can become a health hazard, be a non-profitable use of urban land, or is simply illegal [3,7–10]. At the same time, UA is often seen as a way to increase local food security and provide inclusive employment opportunities for the urban poor [2,3,6,9,11].

However, the contribution of UA to the urban food chain remains difficult to quantify due to the informal nature of the value chain in which UA is positioned [2,4,6]. Even the contribution of UA to the food security of the UA practitioners themselves is difficult to quantify, as UA may be aimed at generating income or diversifying the diet more than directly increasing caloric intake [2]. However, studies have shown that, in most cities in SSA, a considerable portion of the population is involved in UA. A study by Zezza and Tesciotti in 2010 [12] looked at the participation rates of the urban population in UA in 15 developing countries, including four in SSA. In SSA, they found that the average participation rate in UA (growing crops) was 35.5% of the urban population, ranging from 29% in Nigeria to 45% in Malawi [12].

In a recent paper, we argued that to understand and properly value UA, knowledge of prior land use is essential [13]. Prior land use can explain why a site was chosen for UA and how the site fits into the social–economic fabric of the local community. This is exemplified by a recent paper by Robineau and Dugué [14], who showed how diversity in UA could be explained using a socio-geographical approach that includes a historical analysis of the development of UA in Bobo-Dioulasso, Burkina Faso. Expropriation forced farmers to move to the city in search of employment. Finding none, many turned to what they knew in the new environment. Robineau and Dugué [14] classified a number of UA groups based on the origin of their land: (1) land assigned for market gardening during colonial times, (2) remnants of agricultural lands of villages once adjacent to the city, now incorporated, and (3) land protected from development by governments, NGOs, or others and reserved for agricultural activities.

However, prior land use, in addition to influencing the socio-economic position of a particular UA site, also influences the status and potential of its soil with respect to fertility and carbon storage potential. There is a growing body of research on UA and its contribution to urban food security (SDG2), as well as work that focusses on sequestering carbon in soils, including UA soils, to combat climate change (SDG13) [15–18]. However, the potential of UA soils for both remains controversial as there are also studies that show that UA can have no or even a negative impact on the food security of UA practitioners [6] and that agricultural soil can be a carbon source, especially under intensive use [19–21]. These controversies are caused in part by the high variability of the circumstances under which UA takes place. Included in this variability is the past land use and its influence on current practices and results. Good agricultural management on suitable soil may lead to increased carbon sequestration and nutrient content, while poor management and/or an unsuitable soil can lead to high carbon emissions and soil degradation. The knowledge gap with respect to the influence of UA land use history on its present soil status and potential makes it difficult for policy makers and urban planners to consider the value of UA in their city and make decisions that could lead to improved food security for the inhabitants.

Therefore, in this study, we explored the links between prior and current land use and soil nutrient and mineral content, including carbon, at selected urban agriculture sites in Kisumu, Kenya and Ouagadougou, Burkina Faso. The aim was twofold: (1) to further underpin the scientific connection between land use history and current UA practice, and (2) to formulate recommendations on how to incorporate such information for the optimization of the practice to address SDG2 (zero hunger) and SDG13 (climate action). Through this, we aimed to show that urban agriculture, such as that practiced at the studied sites, needs to be valued based on more than its economic aspects. Two case studies were conducted. One was conducted in Kisumu, Kenya, where a previous pilot study indicated a link between current UA soil status and land use history [22], and one was conducted in Ouagadougou, Burkina Faso. An exploration was made of the social and historic contexts of three UA sites in each city, to better understand the reasons for the locations, size, and soil macronutrient content. Second, the soil at the selected sites was described and sampled through the use of soil pits. The soil carbon and nutrient content (nitrogen, potassium, calcium, magnesium, sodium, and manganese), as well as the elemental and mineral composition of the soil, was determined to study the status of the soil at the UA sites in the context of the original parent material of the soil and the land use history. Through this, a narrative was constructed for each site that integrated these points of view. Finally, we considered the current and future value of the preservation of these UA areas.

## 2. Materials and Methods

### 2.1. Study Areas

#### 2.1.1. Kisumu

Kisumu lies on the southwestern border of Kenya with Uganda (Figure 1). Between 1985 and 2012, the population of Kisumu city rose from 175,000 to 367,000 [23]. Locked in between Lake Victoria and the surrounding hillside, arable land is a precious resource in and

around Kisumu, and over 60% of the Kisumu population lives in its crowded slums [10,24]. Unemployment in Kisumu is higher than the national average, at 40% [10,24].

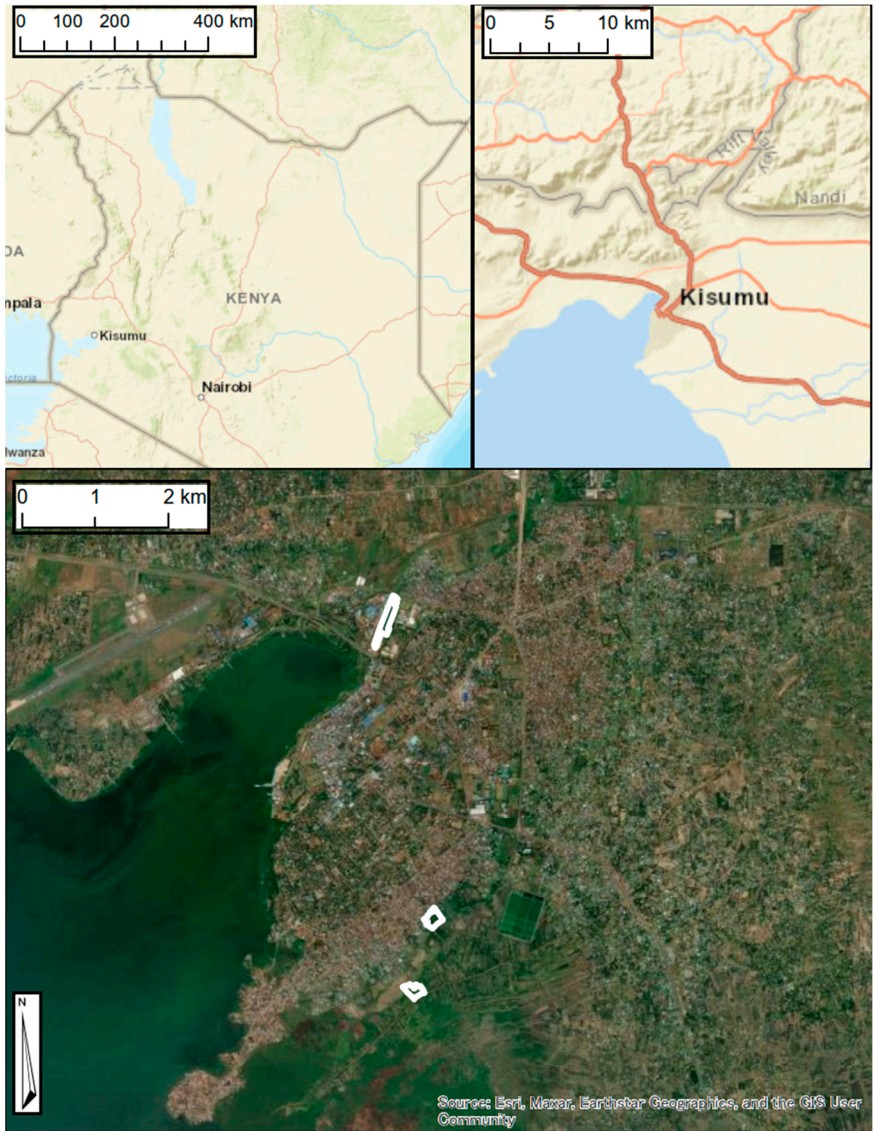

Sources: Esri, HERE, Garmin, USGS, Intermap, INCREMENT P, NRCan, Esri Japan, METI, Esri China (Hong Kong), Esri Korea, Esri (Thailand), NGCC, (c) OpenStreetMap contributors, and the GIS User Community

**Figure 1.** (**top left**) Overview of Kenya with location of Kisumu indicated; (**top right**) overview map of Kisumu; (**bottom**) Kisumu with field sites indicated in white.

Kisumu lies in the sub-humid tropical zone. The yearly average temperature is 22 °C and there are two rainy seasons [25]. The long rainy season lasts from April to June and the short rainy season lasts from November until December [25]. The city lies on Quaternary sediments and Tertiary Volcanic deposits [25]. The parent material of the soils is relatively young, yet deeply weathered soils can be expected in a tropical climate such as in Kisumu. Mean annual rainfall in Kisumu varies from 1200 to 2000 mm [25].

Three of the city's urban agriculture sites were included: the Balaa site, the Nyalenda site, and the Obunga site (Figure 2). The Balaa site lies within a wetland complex and borders the peri-urban/urban boundary of Kisumu just outside the Nyalenda slum area, south of the Nyalenda site. The Nyalenda site lies all along the Nyalenda slum area, which

borders the southern range of Kisumu city. The Obunga site lies to the north of the city center of Kisumu. The site borders the Lake Victoria edge and the railway.

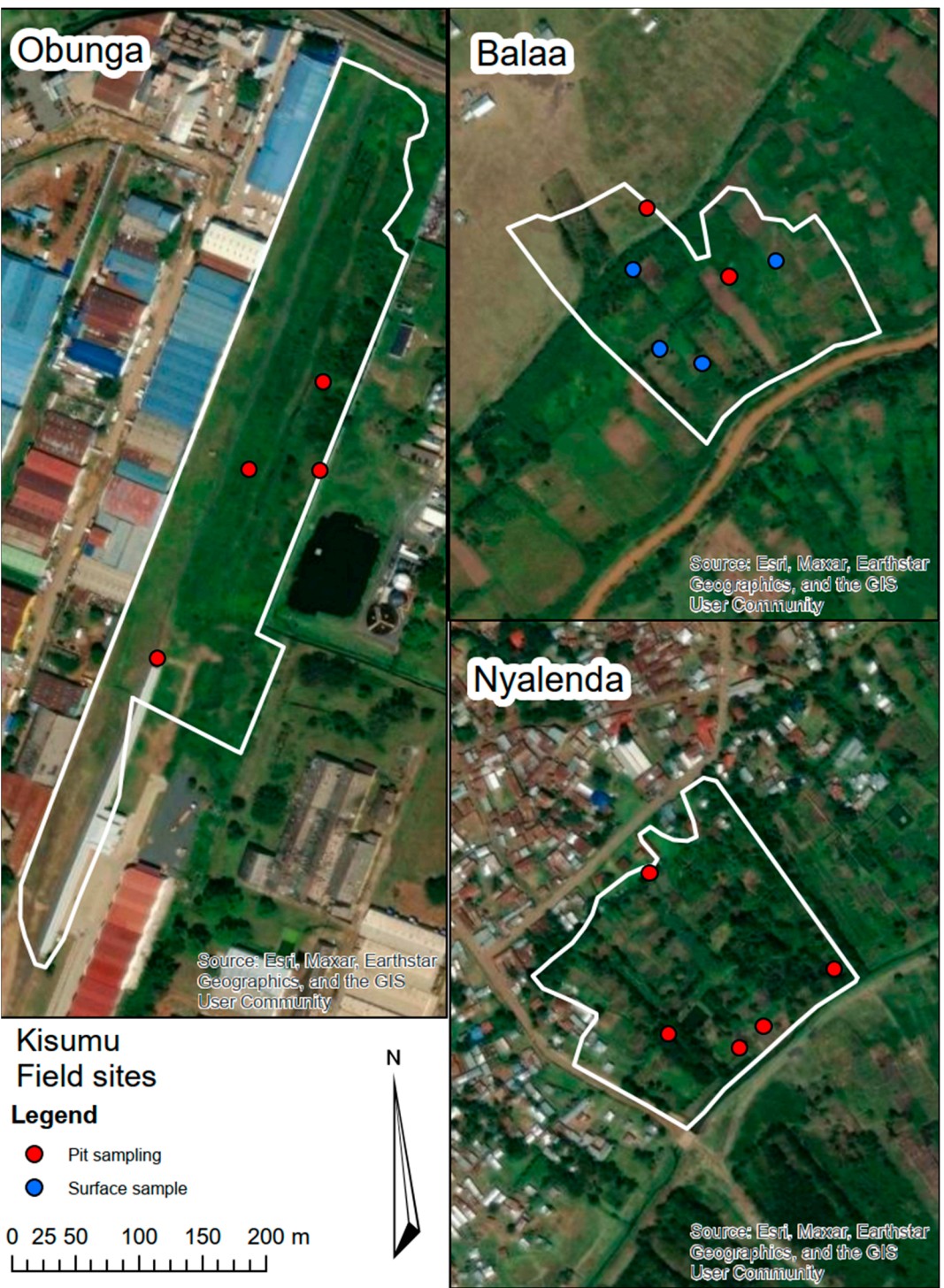

**Figure 2.** Field sites with sampling locations and type indicated. (**left**) Obunga site; (**top right**) Balaa site; (**bottom right**) Nyalenda site. Sampling and observation locations are indicated in red when a soil pit was dug and in blue when only the topsoil was sampled (0–20 cm of the soil).

### 2.1.2. Ouagadougou

Ouagadougou is the capital of Burkina Faso and has approximately 2.5 million inhabitants. Between 1985 and 2012, the population increased from just under half a million

to 2 million [5,26,27]. The climate is characterized as hot semi-arid (Bsh). The annual average temperature is 28 °C, and there is a long dry season from October until April and a rainy season lasting from June until October. Annual average rainfall in Ouagadougou is between 700 and 900 mm [28]. The city lies on the central Mossi plateau (Figure 3). This is part of the West African Craton, which is made up of granites and migmatites of the pre-Birrimian age [27–29]. The area of Ouagadougou is dominated by granites and granitoids containing biotite [30]. Due to the age, soils in and around Ouagadougou formed in highly weathered parent material [28,29]. It is expected that most primary minerals have already been weathered out of the soil. Despite this, research has shown the soils in Ouagadougou to be reasonably rich in nutrients [9]. This presence of nutrients is most likely due to land management, i.e., the application of fertilizers, or due to the location of the main urban agricultural sites next to reservoirs. These reservoirs cause flooding at the UA sites during the rainy season; the material deposited during flooding may be a source of nutrients.

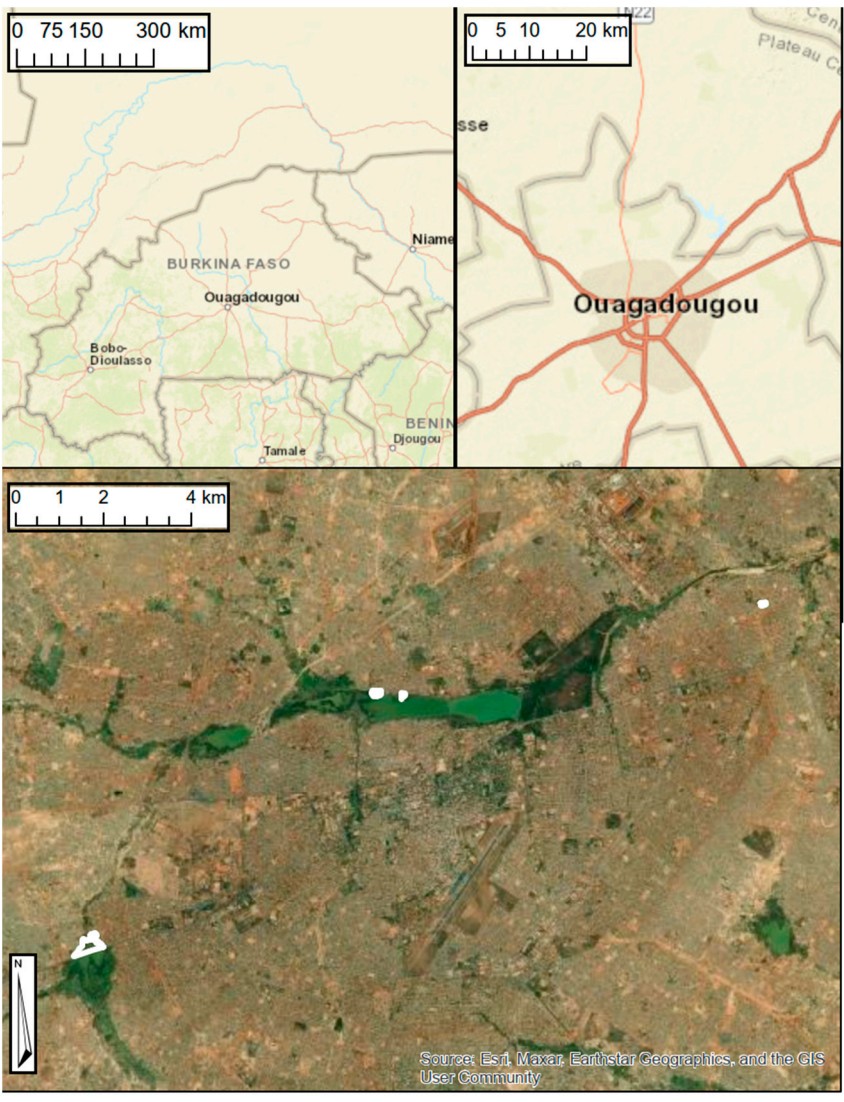

Sources: Esri, HERE, Garmin, USGS, Intermap, INCREMENT P, NRCan, Esri Japan, METI, Esri China (Hong Kong), Esri Korea, Esri (Thailand), NGCC, (c) OpenStreetMap contributors, and the GIS User Community

**Figure 3.** (**top left**) Overview of Burkina Faso with location of Ouagadougou indicated; (**top right**) overview map of Ouagadougou; (**bottom**) Ouagadougou with field sites indicated in white.

Three of Ouagadougou's UA sites were included in the soil survey: the Boulmiougou site, the Tanghin site, and the La Saisonnière site (Figure 4). The Boulmiougou site lies adjacent to a reservoir at the western edge of the city. The Tanghin site is a collection of urban gardens surrounding the central reservoir. The La Saisonnière site is a smaller site in the northeast of the city.

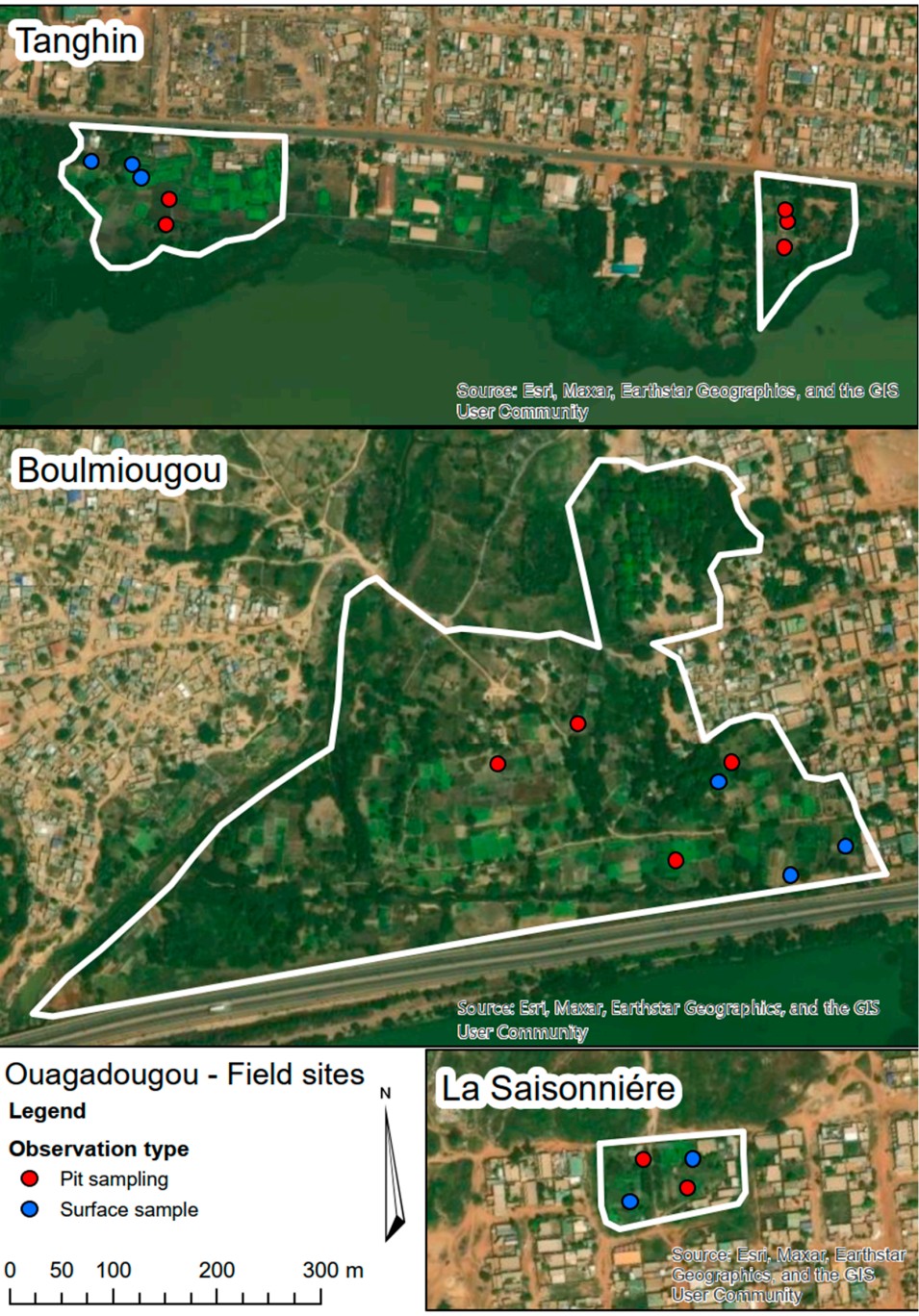

**Figure 4.** Field sites with sampling locations and types indicated. (**top**) Tanghin site; (**middle**) Boulmiougou site; (**bottom right**) La Saisonnière site. Sampling and observation locations are indicated in red when a soil pit was dug and in blue when only the topsoil was sampled (0–20 cm of the soil).

### 2.2. Tracing the Origins of Urban Agriculture in Ouagadougou and Kisumu

In July–August 2016, a survey was done in Kisumu, which was repeated in Ouagadougou in September–October of the same year. After site visits with local researchers and representatives of local NGOs and research institutes (Centre for African Bio-Entrepreneurship and Victoria Institute for Research on Environment and Development in Kisumu, Institut de Recherche en Sciences de la Santé, and the University of Ouagadougou II in Ouagadougou), sites were selected based on their representativeness in terms of size and composition or group structure for UA sites in that city. Each group was approached and asked to participate in the study.

For each of the six sites, a semi-structured interview was held with a chosen representative or chair of the group farming there. The interviews gathered information on the crops grown, common agricultural management practices, and group size and cooperation (see Supplementary Material for interview questions). Additional information on agricultural management was gathered through conversational interviews with 8 farmers while sampling. The location, history, and size of the UA sites were analyzed using information from the interviews and by cross-referencing the information with the historic imagery and literary sources.

Reconstruction of the land use history depended on the age of the site and, related to the site's age, on the availability of (historic) information. The land use history for the younger sites, Balaa (Kisumu) and La Saisonnière (Ouagadougou; <10 years), was readily known by those working at the site and could be confirmed by historic imagery sourced by Google Earth Pro, version 7.3.6. The land use history of older sites was determined using data from the interviews but confirmed using a variety of sources, including historic aerial photography from the African Architecture Matter Collections at the Canadian Centre for Architecture, as well as scientific publications on the cities in question [1,31,32] and popular publications from organizations such as the United Nations [33] on socio-economic development.

### 2.3. Analysis of Soil Nutrient Status at the Urban Vegetable Growing Sites

Concurrent to the interviews, soil pits were dug at each site to evaluate and sample the soil. As the sites were under active cultivation at the time of the survey, sites and sampling methods were selected in conference with the farmers. This was meant to minimize disturbances while maintaining representativeness. At the Tanghin site in Ouagadougou, pits were dug in a gradient parallel to the reservoir as the (partial) flooding of the site is common in the rainy season. A similar method was applied at the Boulmiougou site; at the time of the survey, the west side of the site was not yet in use and still partially flooded due to the recent end of the rainy season. In general, the number of pits depended on the area size of the site. A minimum of 2 pits was dug at the smallest sites, Balaa and La Saisonnière, and up to 6 pits for the largest sites, Boulmiougou and Tanghin (Figure 4). For Balaa and La Saisonnière, between 2 and 4 extra samples were taken from the A horizon (0–20 cm depth) in fields adjacent to those in which the pits were dug, to include spatial variability, as additional soil pits were not feasible. Pits were dug in conference with the farmers at each site so as to minimize disturbance to production and maintain the ability to represent the soil for each site. Pits were dug to a depth of 70 cm or until a C/R horizon was reached. The soil in each pit was described using the World Reference Base (WRB) guidelines for soil description and classified according to the WRB key for soil classification [34]. For each soil horizon, a sample was taken for further analysis; approximately 100 g of material was taken along the depth and width of the horizon in the soil pit.

Gathered samples were processed in the laboratories of the University of Amsterdam by drying them at 60 °C for a period of 24 h and sieving over a 2 mm sieve. Water extracts were prepared (1:2.5) by shaking 20 g of a <2 mm processed soil sample and 50 mL ultrapure water for a period of 16 h, after which the pH and EC of the extracts were determined. $BaCl_2$ extracts obtained by shaking 2 g of soil with 50 mL 0.125 M $BaCl_2$ were analyzed with ICP-OES (Optima 8000 ICP-OES Spectrometer, PerkinElmer, Waltman, MA, USA) to determine

exchangeable nutrient content. A portion of each sample was milled and subsequently used to determine total carbon and nitrogen content with an Vario EL cube CNS analyzer (Elementar, Langenselbold, Germany). To obtain the total elemental composition, samples were analyzed using a NITON$^{TM}$ XL3t-950 He GOLDD+ X-ray fluorescence (XRF) analyzer (Thermo Fisher Scientific, Waltham, MA, USA) under a helium atmosphere in a 2.5 cm Ø sample cup with 6.0 µ polypropylene foil. The XRF was mounted in a test stand and connected to the electricity grid. The XRF filter settings were as follows: mining mode Cu/Z main range (Fe, Zr, Sr, Rb, and heavy metals) 20 s, low range (Ti, Ca, K) 20 s, and light range (Al, Si, P, Mg) 50 s. Reference samples used were USGS GSP-2, GSJ JSL1, NCS DC 71311, NIST 2702, NIST 2780, NCS DC 60117, and LNS CGL006.

Based on the XRF results, a subset of 30 samples was selected for the analysis of the mineral composition by X-ray diffraction (XRD). This was carried out at the Activation Laboratories, Ltd., in Ancaster, ON, Canada. The sample was pulverized and corundum was added as an internal standard in order to determine the amount of amorphous material and poorly crystalline minerals. Amorphous materials and poorly crystalline minerals could not be characterized further using XRD and only a percentage is given per sample for this category as a whole. For the Kisumu samples, the amorphous category mostly contained a smectite-like mineral identified based on broad reflection at about 15–16 Å that shifted to about 17–18 Å after treatment with ethylene glycol. These minerals could not be further classified due to their poorly crystalline structure. For the Ouagadougou samples, these smectite-like minerals were not found. Clay speciation analysis was performed by the gravity settling of particles in a suspension with distilled water. Oriented slides of the <4 µm size fraction were prepared by placing a portion of the suspension onto a glass slide. In order to identify expandable clay minerals, the oriented slides were analyzed air-dry and after treatment with ethylene glycol. The XRD analysis was done on a X'Pert Pro diffractometer with a Cu X-ray source and X'Celerator detector (Malvern Panalytical, Worchestershire, UK)(40 Kv, 40 mA; range 5–70°, 2θ for random specimens and 3–30° 2θ for oriented specimens; step size 0.017° 2θ; time per step 50.165 s; fixed divergence slit, angle 0.5° or 0.250°; sample rotation 1 rev/s). The PDF4/Minerals ICDD database and the X'Pert HighScore plus software 2.0 were used for mineral identification. Specifically, for each site, at least 2 A horizons and 3 C horizons were selected for XRD analysis by determining the greatest variation in the values of Al, Ti, and Fe. Samples were selected to be representative of this variation.

For direct site comparison, results of the soil analysis were averaged per site for the A horizon, as well as for the C horizon directly below. The Pearson correlation coefficient was then determined for the soil mineral composition, the soil elemental composition, and the exchangeable nutrient content for the A horizon and the C horizon using R. The correlation analysis was performed to determine if there was a relationship between soil nutrient content and soil minerals, and especially to examine to what extent the correlations differed between the C horizon and the A horizon. While Pearson correlation analysis is not the most suitable test for the relationships between some minerals and nutrients, the choice was made to use 1 test for all for the sake of the comparability of results.

## 3. Results

### 3.1. Kisumu

#### 3.1.1. Size, Ownership, and Crops at UA Sites

The site characteristics of the UA sites in Kisumu are presented in Table 1. Literature research showed that the age of the UA sites varies, and that both cities have sites where agriculture has been practiced for at least 50 years [31–33,35–37], i.e., the Boulmiougou site in Ouagadougou and the Nyalenda site in Kisumu. The Nyalenda site comprises lands belonging to one group of many that farm on the edge between the Nyalenda slum and the wetland and river complex that it borders (Figure 2). While the site in the study only covers approximately 6 ha, the whole area approaches approximately 22 ha. In an interview with one of the members of the Nyalenda group, the interviewee indicated that

he had worked at the site for his entire adult life (~50 years). At the time of the survey, there were 14 active farmers in the group at Nyalenda. The Obunga site is approximately 3 ha with 30 active farmers, and, at the time of the soil sampling, the Balaa site was 3.5 ha with 10 active farmers working there. The Balaa site has since grown, but it is difficult to determine by how much due to the increased farming activities in the area.

**Table 1.** Kisumu site characteristics, including size, ownership, and data sources.

| Site | Location | Area Size | Site Age | Previous Land Use | Land Ownership | History Confirmation |
|---|---|---|---|---|---|---|
| Nyalenda | South Kisumu | 6.27 ha | +50 years | Agriculture | Inherited, no title deeds | Interview, literary sources [31,33,36] |
| Obunga | Central Kisumu | 3.11 ha | +/−15 years | Semi-natural/industrial | No ownership | Interview, satellite imagery (Google Earth imagery, Kisumu (11 April 2005; 8 July 2019)) |
| Balaa | Southwest Kisumu | 3.56 ha | +/−3 years | Wetland | Owned | Interview, satellite imagery (Google Earth imagery, Kisumu (13 March 2016; 7 August 2016; 8 July 2019)) |

In Nyalenda, the farmers have an ancestral claim to the land (Norman, Nyalenda representative, 2016, Interview by N.T. Jonkman, semi-structured interview, Kisumu, January 17), but no title deeds. The farmers at the Balaa site own the land, whereas the farmers at the Obunga site are tolerated by the Kenya railway, which owns the land (Alex, Obunga representative, 2016, Interview by N.T. Jonkman, semi-structured interview, Kisumu, January 18). The primary crops at all of these UA sites are fresh, leafy green vegetables, for which the demand in the urban areas is high. At the Nyalenda site and the Obunga site, it is a cultivar of *Brassica oleracea*, a type of kale locally known as Sukuma Wiki. At Nyalenda, it is sometimes intercropped with cowpeas (*Vigna unguiculate*), rotated with indigenous vegetables such as spiderplant (*Cleome gynandra*) or African nightshade (*Solanum scabrum*). At the Balaa site, the primary crops grown are such indigenous vegetables. The Balaa site is also the only site with irrigation systems and weighing equipment to standardize sales. At all sites, the soil is ploughed with hand tools to a depth of 30 to 50 cm. An earlier study [22] showed that very few of the urban farmers at the Kisumu sites use inorganic fertilizers, and techniques such as mulching and applying compost are preferred.

3.1.2. Land Use History of UA Sites in Kisumu

The literature confirms that the area around Nyalenda was previously used as agricultural ground in colonial times to function as a buffer zone between the colonial center of Kisumu and the area where the indigenous population was allowed to dwell [31,33,36]. The land of the Obunga UA site is owned by the Kenya railway and the railway runs through the site; the farmer group there maintains the vegetation (Alex, Obunga representative, 2016, Interview by N.T.Jonkman, semi-structured interview, Kisumu, January 18). Google Earth imagery shows the first fields appearing between 2002 and 2005. The Balaa site is the youngest of all six sites in the study and Google Earth imagery shows no activity in March 2016, followed by the large clearance of land for agricultural use in August 2016. Since then, the site has further expanded and the nearby area has similarly developed.

3.1.3. Soil Characteristics in Kisumu

The soil characteristics of the soils of the UA sites in Kisumu are presented in Table 2. Most soils contained clay that showed shrink and swelling behavior matching the description of vertic properties [34]. The A horizon was on average 20–30 cm in depth, which

matches with the depth of the hand-held ploughing tools used (Table 2). The pH ranged from moderately acidic at the Balaa site to neutral at the Obunga site and the Nyalenda site (Table 2). The average CEC of the A horizon was very high at the Nyalenda site, with an average value of 35.9 cmol$_c$ kg$^{-1}$. The CEC at the Obunga and Balaa sites was significantly lower, with a value of 18.8 and 14.8 cmol$_c$ kg$^{-1}$ respectively. The total average carbon content in the A horizon for Nyalenda and Balaa was 28.7 and 28.1 g kg$^{-1}$ soil, respectively, and slightly lower on average at Obunga, at 23.5 g kg$^{-1}$ soil (Table 2).

**Table 2.** Average soil physical and chemical characteristics of the A horizon at the Kisumu research site, with standard deviation indicated in brackets.

| Site | pH | CEC (cmol$_c$/kg) | Total C (g/kg Soil) | Total N (g/kg Soil) | Total Mn (g/kg Soil) | Exchangeable Ca (mg/kg Soil) | Exchangeable Mg (mg/kg Soil) | Exchangeable K (mg/kg Soil) | Exchangeable Na (mg/kg Soil) |
|---|---|---|---|---|---|---|---|---|---|
| Nyalenda (*n* = 5) | 7.34 (0.1) | 35.9 (7.7) | 28.7 (1.0) | 2.2 (0.1) | 9.1 (3.1) | 5935.9 (1434.5) | 529.8 (108.6) | 737.0 (101.4) | 158.4 (86.5) |
| Obunga (*n* = 4) | 6.67 (0.6) | 14.8 (2.9) | 23.5 (0.4) | 2.0 (0.03) | 22.0 (14.1) | 2273.4 (475.9) | 236.8 (82.9) | 569.8 (61.4) | 53.2 (24.4) |
| Balaa (*n* = 6) | 5.74 (0.9) | 18.8 (5.0) | 28.1 (1.2) | 2.6 (0.1) | 4.2 (2.4) | 2788.5 (699.3) | 468.0 (128.8) | 390.8 (251.5) | 118.8 (47.7) |

The plant available or exchangeable nutrients in the Kisumu soils (Table 2) were, according to the FAO standards, generally ranked as being high in the A horizon of the Kisumu sites, especially the amounts of K [38]. The exception to this rule was the Na content for all three sites, which was ranked low to very low, and the Mg content of the soil at the Obunga site, which fell in the medium range [38]. The Obunga site was the least rich of the three sites for all measured cations (Ca, Mg, K, and Na) (Table 2). The Mn content of the soil at all Kisumu sites was very high considering that the background levels are 0.04–1 g/kg soil [39], and there was large variation between the three sites. The Mn content at Obunga was 22 g kg$^{-1}$, while the soil of the A horizon in Nyalenda contained on average 9.1 g kg$^{-1}$ Mn and the soil at the Balaa site 4.2 g kg$^{-1}$ Mn.

3.1.4. Mineral Assemblage of the Kisumu Soils

Figure 5 shows the average results of the XRD analysis for each site in Kisumu. The geology around Kisumu is diverse as it is surrounded by phonolites, with granites to the north, basanites and pyroxenites to the east, and carbonatites to the south. Whereas fast weathering minerals like zeolites, nepheline, and even pyroxenes have been weathered, more common primary minerals like K-feldspar, plagioclase, and anorthoclase are still abundantly present. On the whole, the Nyalenda and Balaa sites were more diverse in soil minerals than the Obunga soil, the latter probably being more influenced by the mineralogy of the nearby granites. With content of 14–15%, quartz was more prevalent in the Obunga soils than in the Nyalenda or the Balaa soils, where there was ca. 10% quartz in the A horizon and ca. 4% in the C horizon. The Nyalenda soil was relatively rich in K-feldspar and anorthoclase, with 16.1% and 8.9% in the A horizon (Figure 5). Furthermore, illite and plagioclase were found in the Nyalenda and Balaa soil, but had no or limited presence (<2–3% of the sample) in the Obunga soil. It should be noted that almost all samples from the Kisumu sites contained considerable amounts of a smectite-like mineral, but, because the mineral was poorly crystalline, these were included in the amorphous category and not separately quantified (Figure 3). It is not possible to state which types of smectite minerals were present in the soil based on their elemental composition.

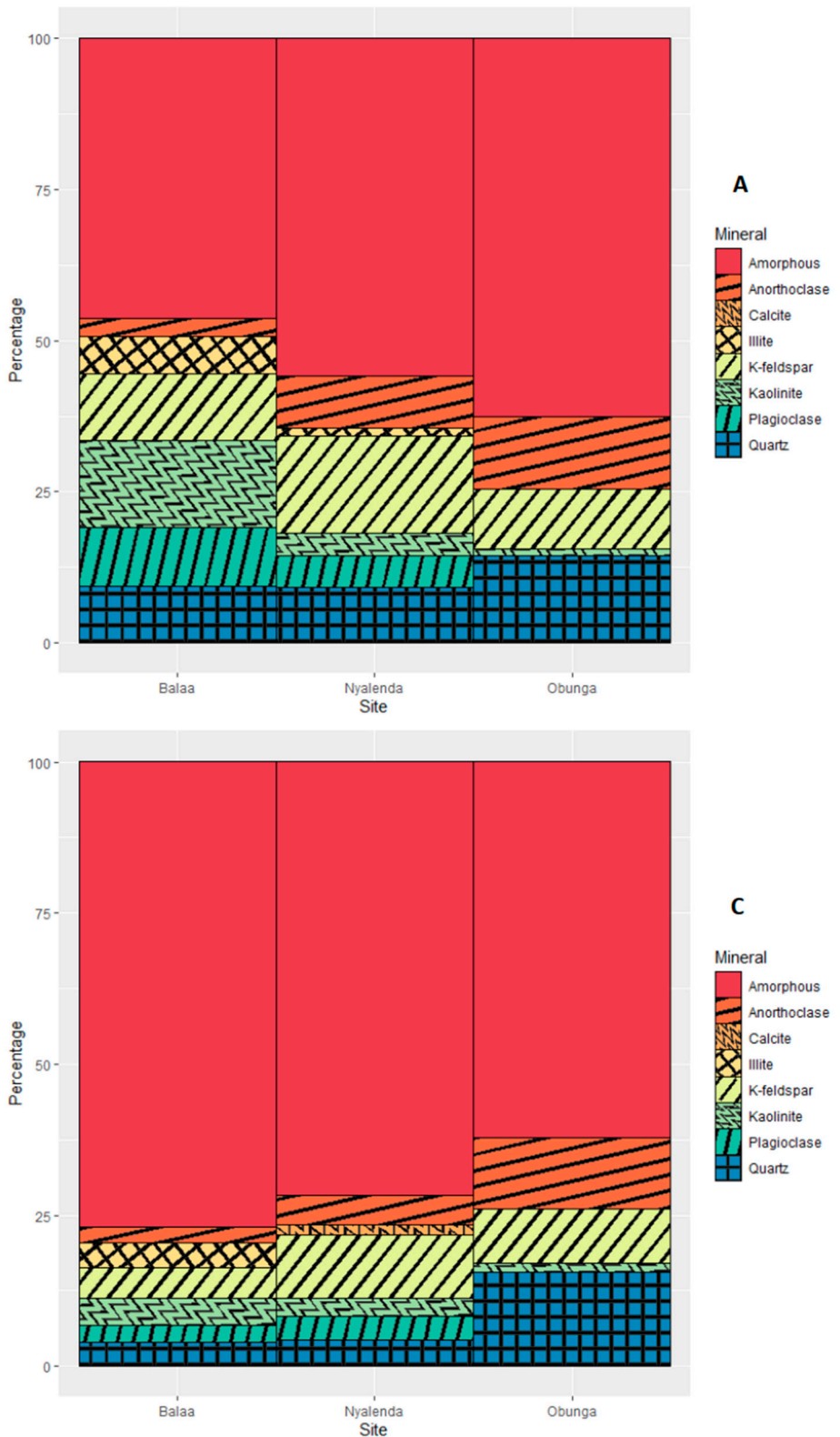

**Figure 5.** Average mineral composition at Kisumu sites in the A (**top**) and C (**bottom**) horizons (% of the dry weight).

### 3.2. Ouagadougou

3.2.1. Size, Ownership, and Crops at UA Sites in Ouagadougou

The site characteristics of the UA sites in Ouagadougou are presented in Table 3. La Saisonnière is the youngest UA site in Ouagadougou in this study. The surface area of La Saisonnière is constrained due to the surrounding infrastructure and it is currently the smallest site at 1 ha. At the time of startup, the neighborhood around La Saisonnière was not developed, but it has since enclosed the site. Whereas, in Kisumu, land availability and suitability are the limiting factors [10,31,33], in Ouagadougou, the first limiting factor is water. This is why the older UA sites, Tanghin and Boulmiougou, are found in proximity to surface water [35]. La Saisonnière is not restricted by this as its practitioners use ground-water pumps. Due to seasonal flooding, the surface areas of Tanghin and Boulmiougou vary throughout the year. Tanghin is approximately 33 ha in size, and Boulmiougou is approximately 18 ha in size.

**Table 3.** Ouagadougou site characteristics including size, ownership, and data sources.

| Site | Location | Area Size | Site Age | Previous Land Use | Land Ownership | History Confirmation |
|---|---|---|---|---|---|---|
| Boulmiougou | Southwest Ouagadougou | 18.21 ha | +50 years | Agriculture | Lease | Interview, literary sources [24,29] |
| Tanghin | North Ouagadougou | 32.77 ha | +50 years | Agriculture | Lease | Interview, aerial photography, literary sources [33,35,37] |
| La Saisonnière | Northeast Ouagadougou | 1.05 ha | ±10 years | Forest stand | Owned | Interview, satellite imagery (Google Earth imagery, Ouagadougou (27 October 2002; 23 February 2008)) |

Tanghin and Boulmiougou are owned by a traditional landowner and the government, respectively. The farmers rent plots of land at these sites (Noungkouni, Tanghin representative, 2016, Interview by N.T. Jonkman, semi-structured interview, Ouagadougou, January 20; Toussaint, Boulmiougou representative, 2016, Interview by N.T. Jonkman, semi-structured interview, Ouagadougou, January 21). At the busiest part of the growing season, there are 55 farmers active at Tanghin. It is not known how many farmers work at the Boulmiougou site. La Saisonnière was founded by a patron as a project for the support and education of Burkinabé women and there are currently 33 active farmers at the site (Yaya, La Saisonnière representative, 2016, Interview by N.T. Jonkman, semi-structured interview, Ouagadougou, October 10). As in Kisumu, the primary crops at Ouagadougou UA sites are leafy green vegetables, primarily lettuce (*Lactuca sativa*) and cabbage (*Brassica oleracea*). The farmers at the Boulmiougou site also use intercropping systems with French beans (*Phaseolus vulgaris*), maize (*Zea mays*), sorghum (*Sorghum bicolor*), and strawberries (*Fragaria* × *ananassa*), among others. At La Saisonnière, some indigenous vegetables and chili peppers are also grown (Yaya, La Saisonnière representative, 2016, Interview by N.T. Jonkman, semi-structured interview, Ouagadougou, October 10). At all sites, the soil is ploughed with hand tools to a depth of 30 to 50 cm. Fertilization schemes differ per farmer per site, but most common is the use of some form of compost or other organic amendments.

3.2.2. Land Use History of the UA Sites in Ouagadougou

In Ouagadougou, the older the site, the more uncertain sources are on the previous land use. In the case of both the Boulmiougou and the Tanghin sites, urban sprawl has caused the urban sphere to encroach on these sites. They now find themselves firmly in the peri-urban to urban continuum, rather than the rural–urban one. The literature shows the Boulmiougou site to be at least 50 years old and the closest housing and urban infrastructure to have been built between 1998 and 2003 [32,35]. For the Tanghin site, the maps show that the surrounding urban infrastructure was also built between 1998 and

2003; furthermore, aerial photography from 1978 shows agricultural activity at the site [37]. La Saisonnière lies in a developing part in the northeast of Ouagadougou city. Google Earth imagery shows that in 2002, the year that the location was founded as a UA site, there was a tree stand at the location and the surrounding area was undeveloped. Recent imagery (2019) shows the site now having little expansion potential due to the development of the surrounding area (Figure 2).

### 3.2.3. Soil Characteristics in Ouagadougou

The soil characteristics of the soils of the UA sites in Ouagadougou are presented in Table 4. The soils of the Ouagadougou sites had a sandy loam or silt loam texture. Some of the deeper soils at the Boulmiougou and Tanghin sites showed a stagnic color pattern, a sign of frequent water stagnation [34]. The depth of the A horizon varied between 10 and 15 cm on average, and the pH was neutral for all three sites, with minor differences (Table 4). The average CEC of the A horizon at the Boulmiougou and Saisonnière sites was similar at 8.3 and 8.6 $cmol_c$ $kg^{-1}$, respectively; the Tanghin site had a slightly higher CEC of 11 $cmol_c$ $kg^{-1}$ on average (Table 4). This pattern is, to some extent, also found for the average total carbon content in the A horizon, with 11.4 g/kg at Boulmiougou, 15.8 g/kg at Saisonnière, and 18.4 g/kg at Tanghin (Table 4).

**Table 4.** Average soil physical and chemical characteristics of the A horizon at the Ouagadougou research sites, with standard deviation indicated in brackets.

| Site | pH | CEC (cmol$_c$/kg) | Total C (g/kg Soil) | Total N (g/kg Soil) | Exchangeable Ca (mg/kg Soil) | Exchangeable Mg (mg/kg Soil) | Exchangeable K (mg/kg Soil) | Exchangeable Na (mg/kg Soil) |
|---|---|---|---|---|---|---|---|---|
| Boulmiougou (*n* = 6) | 7.03 (0.5) | 8.3 (5.6) | 11.4 (5.1) | 1.1 (0.1) | 1347.4 (930.8) | 172.7 (138.1) | 57.5 (48.7) | 40.9 (34.7) |
| Tanghin (*n* = 8) | 6.86 (0.5) | 11.0 (4.9) | 18.4 (0.8) | 1.8 (0.1) | 1710.8 (822.6) | 237.9 (141.6) | 179.8 (173.3) | 134.9 (69.4) |
| La Saisonnière (*n* = 4) | 7.17 (0.3) | 8.6 (4.4) | 15.8 (0.3) | 1.5 (0.1) | 1309.0 (665.4) | 197.8 (76.98) | 179.53 (100.2) | 43.1 (126.1) |

The concentrations of cations in the A horizon of the Ouagadougou sites were lower than at the Kisumu sites but fell within the medium category for sandy soils by FAO standards, with the exception of the average K content at the Boulmiougou site, which was very low [34] (Table 4). The concentrations of cations Ca, Mg, K, and Na at the Tanghin site stood out as they were significantly higher than those at the other two sites (Table 4). In comparison to the Tanghin and La Saisonnière sites, where the K concentrations were ca. 179 mg/kg on average, the concentration of K in the A horizon at Boulmiougou was low, at only 57.5 mg/kg on average (Table 4).

### 3.2.4. Mineral Assemblage of the Ouagadougou Soils

Being derived from a rather homogenous hinterland of granites, granodiorites, and diorites, the soil mineral composition at the Ouagadougou sites (Figure 6) was less diverse than that of the Kisumu sites (Figure 5). All primary minerals, like amphibole, plagioclase, and biotite, were absent, while K-feldspar was only found in trace amounts. Compared to the Kisumu soils, the quartz content was higher, and the smectite-like mineral that was present in the majority of the Kisumu samples was found in none of the Ouagadougou samples. Accordingly, the percentages of amorphous materials as estimated through quantitative XRD analysis were significantly lower in the Ouagadougou samples (Figures 5 and 6). The origin or composition of the amorphous materials of the Ouagadougou is unknown. Whereas plagioclase and anorthoclase were still present in Kisumu soil, plagioclase in Ouagadougou had all been converted to kaolinite. The soils with the highest variability on average in their mineral composition were those at the La Saisonnière site (Figure 6). The soils at the Boulmiougou and Tanghin sites were relatively similar in their total mineral composition. The Boulmiougou site showed a minimal amount of K-feldspar and only had trace amounts of illite, and the Tanghin site soils showed no K-feldspar and a minimal

amount of illite (Figure 6). The amount of amorphous materials at the Ouagadougou sites was higher in the C horizons at all three sites than in their A horizons.

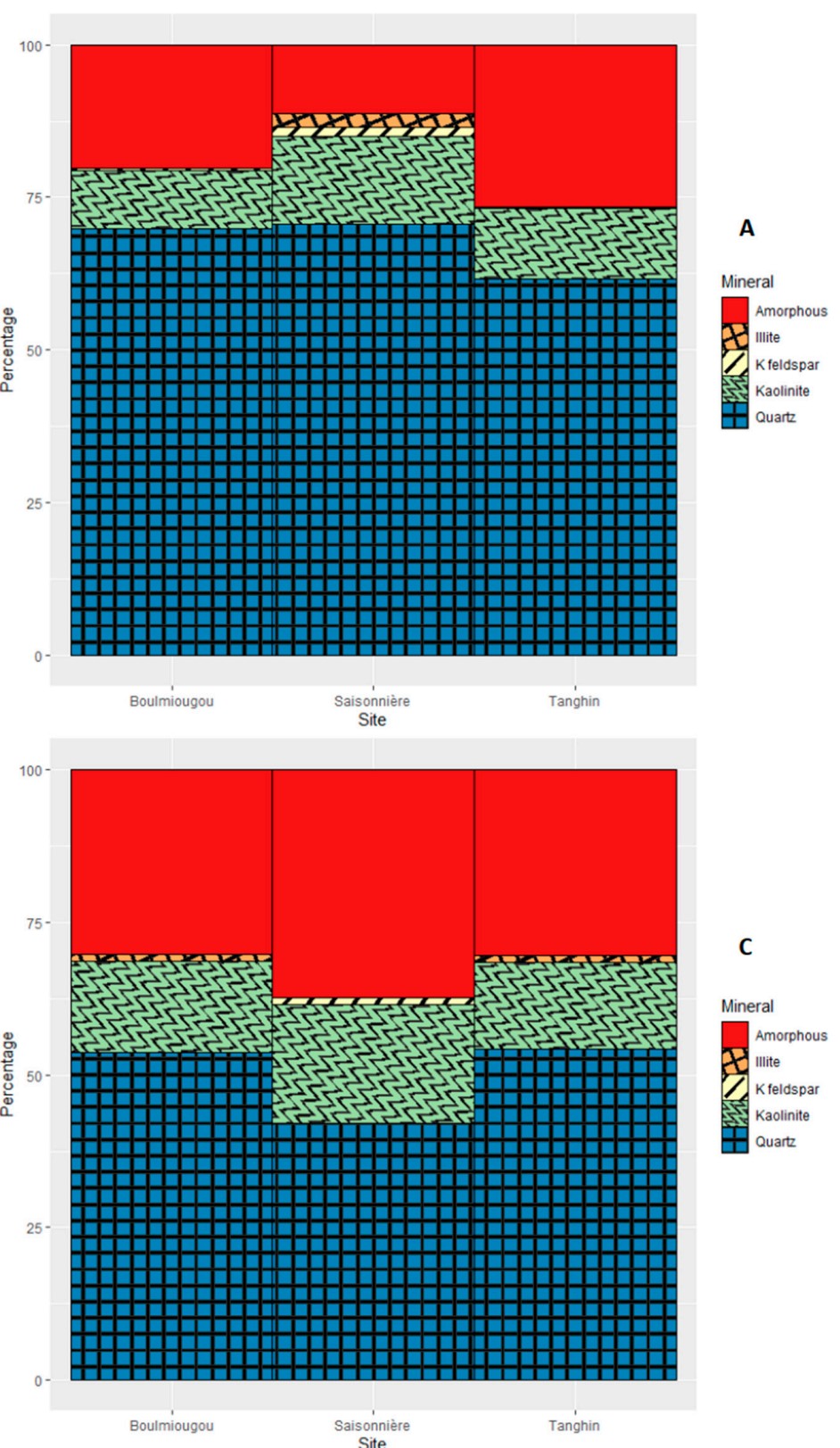

**Figure 6.** Average mineral composition at Ouagadougou sites in the A (**top**) and C (**bottom**) horizons (% of the dry weight).

### 3.3. Relationships between Exchangeable Nutrients and Minerals in the Soil

Pearson correlation coefficients were determined for the samples analyzed by XRD to determine if there were relationships between the minerals and the exchangeable nutrients present in the soil at the UA sites of Kisumu and Ouagadougou together. The results are presented in Figures 7 and 8. For the A horizon, there were a number of significant moderate correlations between soil minerals and exchangeable nutrients (Figure 7), with only one strong correlation—between anorthoclase and K (0.759, sig $p < 0.01$). This stands in contrast to the correlation analysis between minerals from the C horizons and the exchangeable nutrient content. Here, multiple highly significant, strong correlations were found (Figure 8). Most notable was the lack of a correlation between exchangeable nutrients and anorthoclase and illite (2:1) in any of the horizons. None of the minerals were significantly or strongly correlated with Mn either.

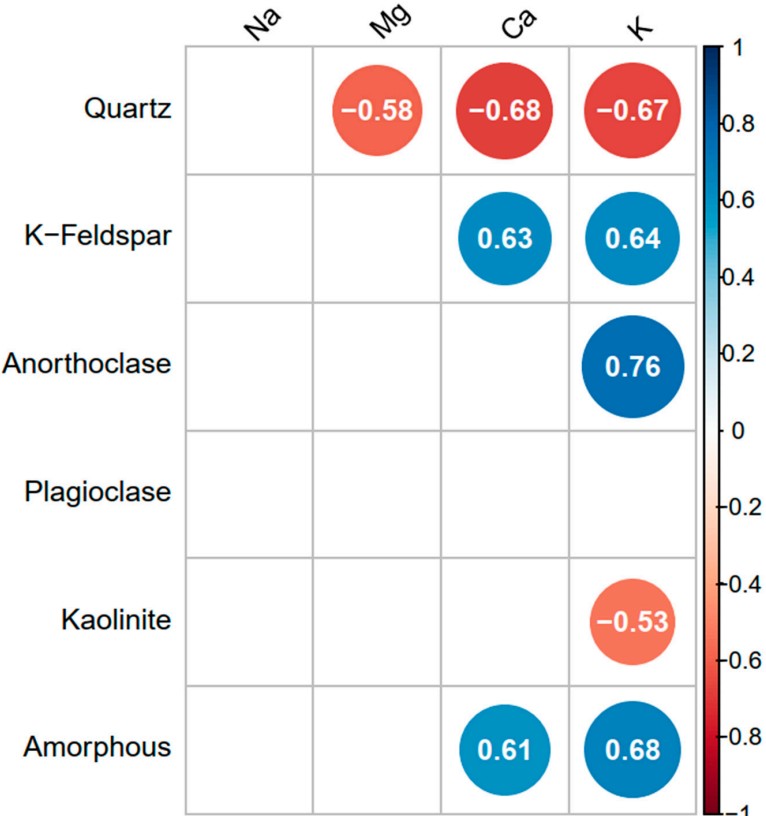

**Figure 7.** Pearson correlation coefficient diagram for the exchangeable nutrients and specific minerals of the A horizon samples. Correlations are only displayed if significant at the 0.05 level.

### 3.4. Relationships between Total Nutrients and Minerals in the Soil

The correlation analysis was repeated for the soil mineral distribution with the elemental composition as determined by XRF analysis. The results are presented in Figures 9 and 10. The highest significant correlation was between K-feldspar and K in the samples of the C horizons (0.958, sig $p < 0.01$) (Figure 10). The correlation between K-feldspar and K in the A horizon was still strong, but less so than for the C horizon samples (0.720, sig $p < 0.01$) (Figure 9). There were more strong correlations between the soil mineral composition and elemental presence in the C horizon, but there were more significant correlations in the A horizon. For example, quartz was significantly negatively correlated to all five elements in the A horizon that were considered (Figure 9), but only significantly negatively correlated to two of the elements in the C horizon that were considered (Figure 10). These correlations were, however, stronger in the C horizon than in the A horizon.

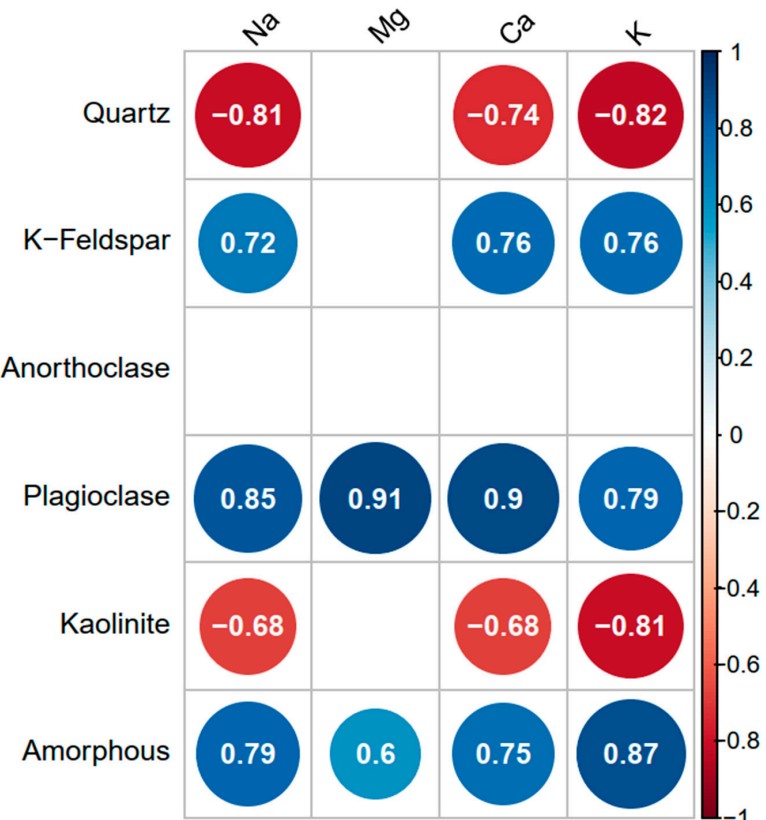

**Figure 8.** Pearson correlation coefficient diagram for the exchangeable nutrients and specific minerals of the C horizon samples. Correlations are only displayed if significant at the 0.05 level. Strength of the correlation indicated through color and size.

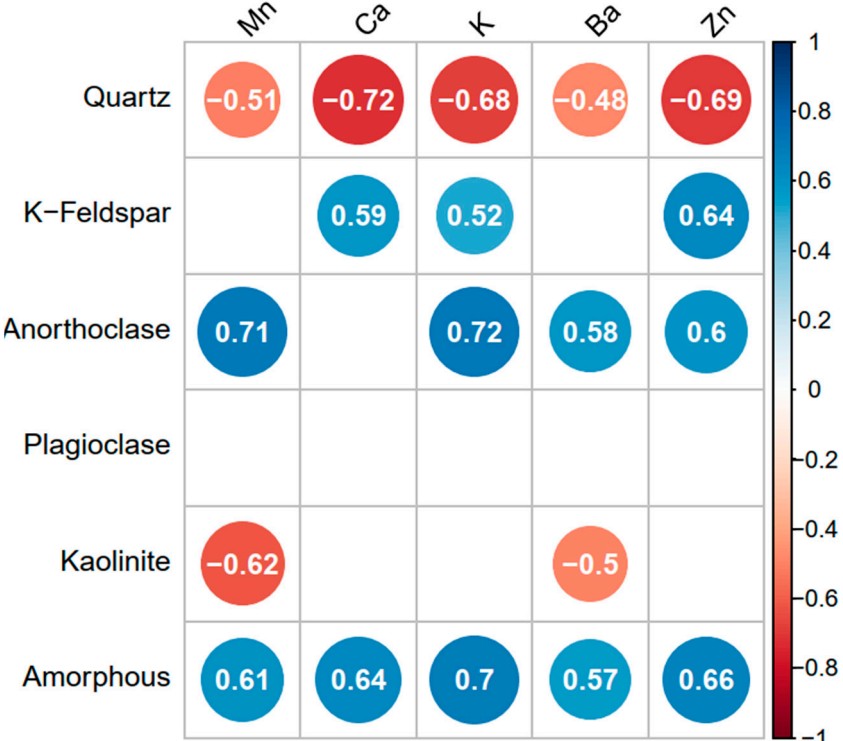

**Figure 9.** Pearson correlation coefficient diagram for the elemental content and soil mineral composition of the A horizon samples. Correlations are only displayed if significant at the 0.05 level.

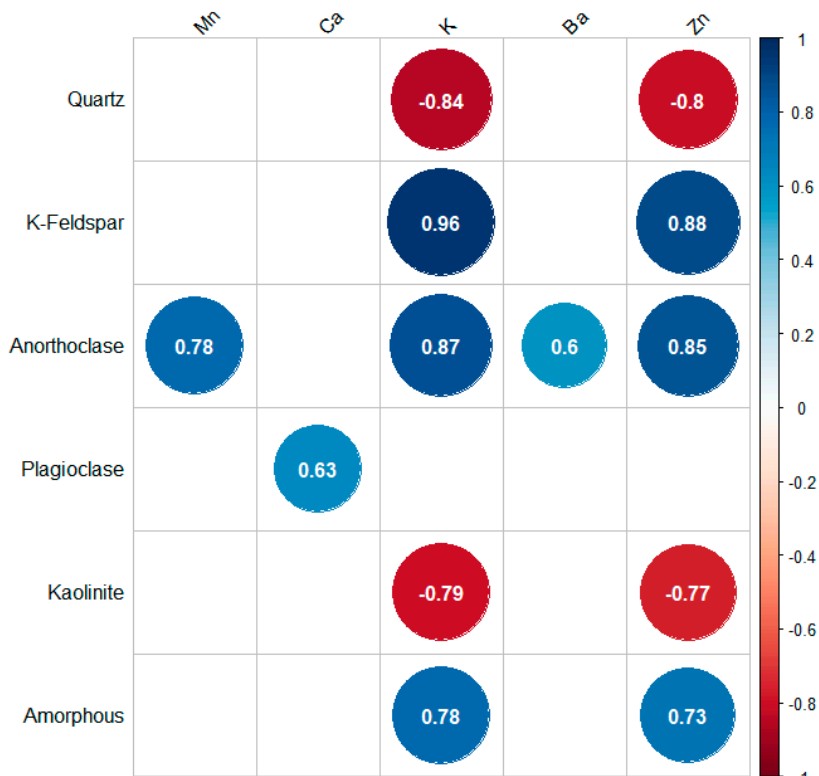

**Figure 10.** Pearson correlation coefficient diagram for the elemental content and soil mineral composition of the C horizon samples. Correlations are only displayed if significant at the 0.05 level.

## 4. Discussion

### 4.1. Land Use History of UA Sites in Kisumu and Ouagadougou

The results of the historical analysis are in agreement with the results from work by Robineau and Dugué [14]: three of the six UA sites, Nyalenda in Kisumu and Boulmiougou and Tanghin in Ouagadougou, are previous agricultural sites. The expanding infrastructure of Kisumu and Ouagadougou has caught up to these sites, changing their surroundings from rural or peri-urban to urban. Obunga deviates from this pattern and resembles the situation often described in the literature regarding UA: people in a precarious position making use of marginal land in the city for agricultural production [19,40,41]. The La Saisonnière and Balaa sites are the youngest of the six sites, and here we see a greater change in land use having occurred in the development of the UA site. In the case of La Saisonnière, there was a tree stand at the location (Google Earth imagery, Ouagadougou, 27 October 2002), and, in the case of Balaa, there was a wetland (Google Earth imagery, Kisumu, 13 March 2016).

### 4.2. UA Site Choice and Soil Status

For all six sites, including the Obunga site, a case can be made that the locations were consciously selected based on environmental factors.

- The Tanghin and Boulmiougou sites lie adjacent to some of the only open water bodies in and around Ouagadougou, which is the primary limiting factor for agriculture in this area of Burkina Faso. Seasonal flooding may have also offered improved soil conditions at these locations in contrast with others, as the soil shows the impact of this in the presence of stagnic color patterns.
- Agriculture at the Nyalenda site was actively encouraged during colonial times [31], and the soils there are locally known as black cotton soils, which are well suited for agriculture (Azeal, Balaa representative, 2016, Interview by N.T.Jonkman, semi-structured interview, Kisumu, January 18).

- Since the La Saisonnière site is located at a former tree stand, it is not unlikely that the soil there contained more organic matter and nutrients than the soil of the surrounding area as a consequence.
- The Obunga site may not seem an ideal location, surrounded by industrial sites and along a rail line, but the soil at the location shows similar properties to that at the Nyalenda site based on observation.
- The same soil as at the Obunga site is also present at the Balaa site and it is the inhospitable surroundings and lack of infrastructure that have so far limited the development of the area. It is likely that the increased development of infrastructure around the area due to Kisumu's expansion has improved the accessibility of the Balaa site and helped to spur activity in the area.

The likelihood that all the UA sites studied were chosen for their soil is supported by the results of the soil analyses. Despite the high standard deviations overall, the analysis of the soils at the UA sites by themselves show soils rich in nutrients in Kisumu, especially in regard to carbon and potassium, with the Nyalenda site soil standing out as being particularly rich in terms of exchangeable cations like $K^+$ and $Ca^{2+}$ in the soil (Table 2). In Ouagadougou, the soils of the UA sites generally fall within the medium range in terms of nutrient content (Table 4); this can be considered good agricultural soil as more sandy soils often have lower nutrient content [35]. Amongst the Ouagadougou sites, the Tanghin site stands out, with the highest content of carbon and nitrogen, as well as exchangeable cations such as calcium (Table 4). This may be due to seasonal flooding from the reservoir or local soil management practices, which include mulching and applying compost to add organic material to the soil.

Our findings support and further strengthen the conclusions from our previous study [13], i.e., that the prevalent notion that UA sites are generally founded on marginal lands, where the initiators had little or no influence on the selection of the location, must be revisited. Instead, the land use history, including the conscious selection of UA sites based on favorable (soil) properties to practice agriculture when the sites were first founded, should explicitly be considered when studying UA. In such an assessment, present-day soil characteristics can provide important insights.

A further example of the value of soil characteristics in providing the context of the land use history of UA sites is provided by the Mn content of the soil analyzed from the Kisumu sites. At Balaa, the Mn content of the soil is the lowest measured in Kisumu, being 4.2 g kg$^{-1}$ soil on average (Table 2). At Nyalenda, this increases to 9.1 g kg$^{-1}$ soil, and it spikes at Obunga, with average content of 22.0 g kg$^{-1}$ soil (Table 2). As such, the Mn levels at Nyalenda, and especially Obunga, are far above what is considered a background level, which is between 0.04 and 1.0 g kg$^{-1}$ soil [36]. A measurement error is unlikely as the samples from Ouagadougou and Kisumu were analyzed in one batch, ordered randomly, and the XRF was calibrated using standards every 5 to 10 samples. Mn is present in many different types of minerals, including biotites and amphiboles, but this generally does not cause concentrations above the stated background levels in the soil. Anthropogenic sources of Mn include wastewater and sewage sludge; emissions from alloy, iron, and steel production; combustion from fossil fuels; and mining and mineral processing [42]. The Obunga site is surrounded by industry and the site itself is adjacent to a railway line. Kisumu is known as the terminus of the Kenya railway and is an important link in trade with Uganda [36]. These industrial activities may have caused the increased Mn to leach into the environment over time.

### 4.3. Soil Nutrient Status: Site Management and History

The correlation analysis shows that the nutrient content of the soil, both exchangeable and total, is correlated to some degree to the soil mineral composition, as was expected. For the exchangeable nutrients, it is especially noticeable that there are more strong correlations between soil minerals and nutrient content for the C horizon than for the A horizon (Figure 5). This is not surprising and is a clear indication that the nutrient content in the

A horizon is influenced by agricultural management, and that factors such as soil organic matter have superimposed the influence of soil minerals. As mulching, composting, and intercropping are the preferred management practices at the studied UA sites, organic matter is regularly added to the soil. An earlier study at the Nyalenda UA site in Kisumu also showed that land management has a considerable influence on soil nutrient content [13].

However, the correlation analysis between the mineral composition of the A horizon and the total nutrient content shows a more diffuse pattern. While the strongest correlations were observed in the C horizons, a larger overall number of significant correlations was observed in the A horizons (Figures 7 and 9). Together, these results imply that, notwithstanding the important influence of soil management, the original parent material and fertility status, as directly determined by the historic selection of the site, continue to exert an important influence on the total pool of nutrients available today.

Furthermore, not to be discounted is the pool of amorphous materials found in the UA soils. For the Kisumu samples, the amorphous materials could largely be identified based on clay speciation analysis as being a smectite-like mineral, which could not be properly classified as such due to the poorly crystalline structure. For the Ouagadougou soils, this smectite-like mineral or similar was not found. With 15 to 35% of the soil minerals of the Ouagadougou soil being these amorphous materials, whose origin and composition is unknown, this could represent a considerable influence on agricultural production at the studied UA sites. More research will have to be done to determine the origin and effects of these amorphous materials. Possible sources could be the seasonal flooding, mostly affecting the Tanghin and Boulmiougou sites, or the input of soil amendments from the farmers through their fertilization practices.

### 4.4. Implications for UA Soil Management

Our study shows that the current UA sites were and are not necessarily initiated on marginal, nutrient poor wastelands, but can also originate on nutrient-rich parent materials that were consciously chosen for this reason and that, in spite of decades of soil management, continue to exert an important influence on the soil nutrient status. This has important implications for the direction of future soil management in UA. With UA soils that are inherently more fertile than thought and the standard management advice generally only based on maize production or other grains, rather than the leafy vegetables favored in UA, unsuitable advice for the use of inorganic fertilizer may have adverse effects [43,44]. Unsuitable fertilizer advice stimulates UA practitioners, often already in a precarious socio-economic position, to waste valuable resources and might ultimately even lead to over-fertilization, which can adversely affect the soils and eventually their crop yields [45–47]. Instead, the fertility status of an UA site should be assessed within the context of the history of the site and be based on a thorough mineralogical assessment to assess the weatherable reservoir of (micro)nutrients present.

Soil deficiencies for different crops may include a lack of (available) N, P, and K, but also other specific micronutrients, such as boron. Many inorganic fertilizers are not capable of addressing these macronutrient deficiencies, and novel soil amendments such as rock dusts may be necessary. Furthermore, the traditional practices of the UA practitioners should also be taken into consideration—for example, the growing of local species, such as African nightshade (*Solanum scabrum*) in Kisumu, or intercropping systems, such as those used at the Boulmiougou site in Ouagadougou. More traditional vegetables may be more adapted to the local conditions and soil, leading to the more efficient use of resources and better soil conservation through this. A more socio-economic approach to constructing fertilizer advice may lead to new insights for soil management and is more likely to be adopted by practitioners in a sustainable fashion. Research has shown that UA practitioners base their management strategies on more than merely optimized yields and that they are also unlikely to adopt practices that do not take into account their socio-economic circumstances [13]. This should then subsequently be translated into a bespoke

management strategy including a fertilization scheme that addresses deficiencies specific to a particular site, as well as taking into account the means and methods of the farmer.

*4.5. SD2 and SDG13 in the Context of UA Soils*

The context-specific variations in the inherent soil mineralogical composition of UA soils also has important implications for the current optimization of other ecosystem services besides providing food crops alone. Most prominent in this respect is the increasing aspiration to maintain or even increase soil carbon levels in view of SDG13 (climate change). In UA, this may be a double-edged sword. Agricultural management has the potential to either create a carbon source or a sink. Further optimization of UA in favor of SDG13 could potentially be achieved through investing in climate-smart agricultural practices at UA sites. Sites in Kisumu might be suitable for the adoption of agroforestry. Other practices, such as no-tillage agriculture or better irrigation mechanisms, could be used to support UA at sites such as those in Ouagadougou. This would be advantageous not only in the case of SDG13, by maintaining or increasing the soil carbon stock, but would also support SDG2 (zero hunger), by maintaining or increasing the local food supply and availability. The two most prominent mechanisms of C sequestration in soils are (micro)aggregation and adsorption on mineral surfaces [48]. Both mechanisms strongly depend on the soil mineralogy. Novel soil amendments such as locally sourced rock dusts and the adoption of no-tillage agriculture could potentially be used to increase C sequestration through these mechanisms by increasing the availability of weatherable minerals and conserving natural aggregation in the soil [49–51]. The conclusion that UA sites in SSA can be much richer in weatherable minerals than often thought also means that they hold much larger potential to sequester C via the mentioned mechanisms. The presence of these materials means that there is greater potential for the sorption of organic matter on mineral surfaces.

## 5. Conclusions

The historic analysis of the UA sites in this study shows that most site locations were likely consciously chosen for the presence of resources like water and soil of sufficient quality to practice agriculture. The analysis of soil at the sites shows that they generally contain medium to high concentrations of exchangeable cations, carbon, and nitrogen. Though the lack of quantification remains, these UA sites do deliver leafy vegetables with a reduced chance of spoilage due to reduced transportation distances. Locally produced food can make a city more resilient as it becomes less dependent on infrastructure and transportation. This all contributes to SDG2—zero hunger. The age of these sites and their continuing survival can be seen as a consequence of the difficulty of founding permanent structures in places that often flood, but they are also a sign of their integration within the local food chain. Established sites such as Nyalenda or Boulmiougou likely play an important role in local food chains. In the interest of furthering SDG2, it would be worthwhile to further examine these networks to understand where infrastructure improvements could help to increase local food and nutrition security.

This is also why it can be valuable for policymakers to carefully consider UA and the location and history of UA sites. Well-established sites where primary infrastructure and knowledge is already present can be optimized more easily with small investments into, for example, storage facilities. This can lead to increased income for UA practitioners and increased access to and availability of food locally. In the context of furthering SDG13, more in-depth studies of the soil at these UA sites would be needed to quantify their potential for further C storage in the soil. Simultaneously, fertilizer studies done in partnership with UA practitioners may lead to more sustainable fertilization schemes, which take into account not only soil health and crop yields but also the UA practitioners' knowledge, priorities, and means.

**Supplementary Materials:** Supporting data on soils can be found at https://data.4tu.nl/private_datasets/3mop0eaekFuoHNq5_-YaEWCXCkdQeW9DdPHKBJ5bZg4 access from 29 August 2023.

**Author Contributions:** N.T.J., B.J. and K.K. contributed to the study design. N.T.J. conducted the soil field survey, interviews, and laboratory analysis of samples and contributed to the data analysis. H.B. contributed to the background data on minerology and the mineralogical analysis. K.K. and B.J. contributed to the data analysis and background data on soil and social sciences. The first draft of the manuscript was written by N.T.J. and all authors commented on previous versions of the manuscript. All authors have read and agreed to the published version of the manuscript.

**Funding:** This research has been supported by the Nederlandse Organisatie voor Wetenschappelijk Onderzoek (grant no. W 08.250.200).

**Data Availability Statement:** Data is contained within the article and Supplementary Material.

**Acknowledgments:** We would like to sincerely thank all those who aided us in the preparation and execution of this research project, specifically the Kisumu VIRED team, including Joash Barack Okeyo-Owuor and Dan Abuto; the CABE team in Nairobi, including Hannington Odame; the Ouagadougou IRSS team, including Sylvin Ouedraogo; and the Netherlands organization for scientific research (NWO-WOTRO).

**Conflicts of Interest:** The authors declare no conflict of interest. The funders had no role in the design of the study; in the collection, analyses, or interpretation of data; in the writing of the manuscript; or in the decision to publish the results.

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
