# Peer review of "Site History’s Role in Urban Agriculture: A Case Study in Kisumu, Kenya, and Ouagadougou, Burkina Faso"

_land, doi:10.3390/land12112056_

Round 1
Reviewer 1 Report
Comments and Suggestions for Authors
The supplemental materials for the manuscript were not available to me to review. For that reason, it is difficult for me to gauge whether or not interview questions about "group composition and internal relationships (Lines 158-159) constitute human subjects research. No mention is made of compliance with IRB or similar protocols for human subjects research (in the manuscript itself or in the end matter- author contributions, acknowledgements etc.)
Otherwise, this is a well written manuscript covering and interesting subject that is just now receiving more attention.
Author Response
Thank you for your review of our manuscript, we have responded in the submitted file

Reviewer 2 Report
Comments and Suggestions for Authors
The manuscript addresses the interesting subject of urban agriculture in sub-Saharan Africa. The article provides interesting information, such as the fact that, contrary to what is often assumed, the soils devoted to urban agriculture are not necessarily marginal soils and, in the cases studied, they have been chosen for their relatively good quality.
However, some revisions would help improve the article. I miss some information on certain management practices possibly carried out by farmers (fertilisation, manuring, liming…). Some results are barely commented on in the Discussion (relationship between available nutrients and mineralogical composition, relevance of non-crystalline materials, …). Detailed comments are included in the annotated manuscript.

Comments on the Quality of English LanguageMinor editing is needed.
Author Response

(The authors gave the same response as above.)

Reviewer 3 Report
Comments and Suggestions for Authors
Recommendations to authors
General comments:
The idea of studying soils for urban agriculture is interesting but not new, so there is an issue regarding the novelty of this study; I think the aim of the study is not clear enough. Therefore, I think you might need to make several modifications and improvements (mainly in introduction/discussion/conclusions) to clearly present what new this study offers and how the findings of current research can be useful further.
My main concerns are the following:
[1] The purpose of the current study is not clear, although the idea of assessing the soil quality/management for agricultural use over time is interesting, the aim of the study is vague. The conclusion that “most site locations were likely consciously chosen for the presence of resources like water and soil of sufficient quality to practice agriculture” is interesting but rather expected, and it does not offer something new with the only exception maybe that these soils may not need excessive fertilization.
[2] Information about the sampling design is missing.
[3] More clear suggestions based on the findings of this work would be a great addition; I think that authors can either reformulate and rewrite the subchapter “Implications for UA soil management” or they can simply add a new section for their suggestions/recommendations for “farmers” for the future management of UA soils in the context of adopting a more sustainable or socio-economic approach, such as the controlled use of fertilizers (because soils might have already a high soil nutrient level), optimization of the agricultural practices based on socio-economic criteria, or even propose different uses of UA lands such as community gardens (establish community gardens in vacant lots or shared spaces where residents can collectively grow food that promote community engagement and food security), use of IoT devices that can monitor environmental conditions and automate tasks to ensure the quality of the products etc.
[4] The format of the tables is not compatible with the template provided by the journal and the instructions to authors. Some figures have an unusual “color noise” (unnecessary use of color), and those that refer to “average mineral composition” use pie graphs instead of stacked bars/columns, while data (namely percentages) are missing. Please remove the color information and maybe consider using simple (bars/stacked 100% column) graphs.
Detailed comments:
- Line 2: I think the title can be improved.
- Line 35: Improve English.
- Line 144, Figure 4: the sampling design seems to be unbalanced since a large area of Boulmiougou has not been sampled. Please explain. In all cases the samples/measurements taken are below five; given the fact that these soils can be considered as “Technosols”, there is a high possibility that some areas might have over 20% addition of anthropogenic materials or other continuous impermeable technic hard materials. Do you believe that this low number of samples can be representative of the given area? In the Excel spreadsheet it seems that the total number of samples is greater than in the paper or than those presented in figures. Please add the information needed, if missing.
- Line 157: You must provide at least the following information about the interviews: (a) sample size (how it was calculated and if it is representative of the given population); (b) the type of the questionnaire (what was the type of the questions used: closed-ended; open-ended; semi-structured; ranking; dichotomous; multiple response or other?)
- Line 159: Where is the file for the supplementary material to check the interview questions?
- Line 200: “a subset of samples was selected” you must add how many samples.
- Line 202: Better replace “The sample” with “Each sample”.
- Line 235: Regarding the “sampling locations”, no information is given on how these locations were selected and how the sampling design was set (was it simple random, stratified, systematic, cluster or other? You must provide this info in the text and explain why this sampling design was selected).
- Line 263, Table 1: the format of the table is not compatible with the guidelines given by the journal.
- Line 290, Table 2: the format of the table is not compatible with the guidelines given by the journal.
- Line 512, Figure 5: the figure is about the “Average mineral composition at Kisumu sites in the A and C horizons (% of the dry 312 weight)" but surprisingly no quantitative information is given even with the form of percentages. I would prefer you to remove the pies and use stacked % column charts. The symbols used are inappropriate and the color information is decorative and not informative.
- Line 353, Table 3: the format of the table is not compatible with the guidelines given by the journal.
- Line 380, Table 4: the format of the table is not compatible with the guidelines given by the journal.
- Line 402, Figure 6:
- Line 417, Figure 7: the index is missing; although it is mentioned that “Strength of the correlation indicated through color and size” you must provide a scale and a corresponding index for the figure.
- Line 438, Figure 8: same comment as above.
- Line 498: Not “clues” but “insights”.
Author Response

(The authors gave the same response as above.)

Round 2
Reviewer 2 Report
Comments and Suggestions for Authors
Although the manuscript has been somewhat improved, some of my main concerns have not been satisfactorily addressed. My major objections are related to (1) the highly weathered character of Ouagadougou soils, evolved from granitic rocks in a semiarid climate; (2) the presence in these and other soils of amorphous materials, whose identification and quantification methodology is not clear, which is contradictory with the highly weathered character (amorphous materials occur in young soils). Therefore, my recommendation remains “major revisions”.
Detailed comments are included in the annotated manuscript.

Comments on the Quality of English LanguageMinor editing required
Reviewer 3 Report
Comments and Suggestions for Authors
Recommendations to authors
The format of the tables is still not compatible with the template provided by the journal and the instructions to authors. Check the template of the Journal: https://www.mdpi.com/files/word-templates/land-template.dot. You can replace the colors in Figures 5 and 6 with patterns.
- Line 2: I think the title can be improved further. Consider the following title: “Site History's Role in Urban Agriculture: A Comparative Study in Kisumu, Kenya, and Ouagadougou, Burkina Faso”.
- Line 294: Check the format of table 1.
- Line 321: Check the format of table 2.
- Line 346: You can replace colors in Figure 5 with patterns.
- Line 398: Check the format of table 3.
- Line 425: Check the format of table 4.
- Line 450: You can replace colors in Figure 6 with patterns.
